# Influence of Natural Polysaccharides on Intestinal Microbiota in Inflammatory Bowel Diseases: An Overview

**DOI:** 10.3390/foods11081084

**Published:** 2022-04-09

**Authors:** Qi Li, Linyan Li, Qiqiong Li, Junqiao Wang, Shaoping Nie, Mingyong Xie

**Affiliations:** 1State Key Laboratory of Food Science and Technology, China-Canada Joint Lab of Food Science and Technology (Nanchang), Key Laboratory of Bioactive Polysaccharides of Jiangxi Province, Nanchang University, Nanchang 330047, China; 15835563385@163.com (Q.L.); lealinyan@163.com (L.L.); spnie@ncu.edu.cn (S.N.); myxie@ncu.edu.cn (M.X.); 2Center for Microbial Ecology and Technology (CMET), Faculty of Bioscience Engineering, Ghent University, 9000 Ghent, Belgium; qiqiong.li@ugent.be

**Keywords:** polysaccharides, inflammatory bowel disease, intestinal microbiota, intestinal immunity, short-chain fatty acids

## Abstract

The incidence of inflammatory bowel disease (IBD) has increased in recent years. Considering the potential side effects of conventional drugs, safe and efficient treatment methods for IBD are required urgently. Natural polysaccharides (NPs) have attracted considerable attention as potential therapeutic agents for IBD owing to their high efficiency, low toxicity, and wide range of biological activities. Intestinal microbiota and their fermentative products, mainly short-chain fatty acids (SCFAs), are thought to mediate the effect of NPs in IBDs. This review explores the beneficial effects of NPs on IBD, with a special focus on the role of intestinal microbes. Intestinal microbiota exert alleviation effects via various mechanisms, such as increasing the intestinal immunity, anti-inflammatory activities, and intestinal barrier protection via microbiota-dependent and microbiota-independent strategies. The aim of this paper was to document evidence of NP–intestinal microbiota-associated IBD prevention, which would be helpful for guidance in the treatment and management of IBD.

## 1. Introduction

Inflammatory bowel disease (IBD), including ulcerative colitis (UC) and Crohn’s disease (CD), is a multifactorial disease characterized by immune system relapse and inflammation of the gastrointestinal tract. An epidemiological study in 2017 reported that more than 6.8 million people suffered from IBD globally [1,2,3]. Although the etiology and pathogenesis of IBD are currently unclear, studies have reported that IBD is linked to the environment, microorganisms, and immune mediators of genetically susceptible hosts [4]. Conventional drugs for IBD patients mainly include anti-inflammatory drugs such as 5-aminosalicylic acid (sulfasalazine, olsalazine, and mesalazine), immunosuppressive drugs such as glucocorticoids (prednisolone and beclomethasone dipropionate), azathioprine, and 6-mercaptopurine, and biological products, such as infliximab, adalimumab, and golimumab [5]. Although those drugs are available for use in IBD treatment, they are associated with a range of side effects, such as weakened immunity. In addition, the major challenge encountered in IBD treatment is the limited initial reactivity and the high costs of biological products. Therefore, many IBD patients are still waiting for more effective and safer alternatives [6].

Natural polysaccharides are long-chain polymers formed by more than 10-monosaccharide molecules joined together by a glycoside bond, which can be divided into lipopolysaccharides and heteropolysaccharides [7]. Based on compositional monosaccharide, they can also be classified into homopolysaccharides and heteropolysaccharides. Homopolysaccharides are a polymer composed of the same monosaccharide, such as starch, β-glucan, galactan, etc. Heteropolysaccharides are polymers composed of two or more monosaccharides, such as glucomannan, arabinogalactan, pectin, and so on [7]. They are widely distributed in natural resources, including plants, animals, fungi, bacteria, and algae. NPs are among the essential biological macromolecular substances required to maintain the everyday activities of organisms [8]. In recent decades, researchers have gradually recognized the biological roles of NPs, including hypoglycemic [9,10], hypolipidemic [11], antitumor [12], antioxidant [13], immune modulatory [14], and anti-inflammatory activities [15]. For example, β-glucans *Lentinus*, *Cordyceps,* and *Ganoderma* species were shown to have potent immunomodulatory, antitumor, and antivirus effects [16]. Arabinoxylan [17,18], glucomannan [19], pectin [20], galactomannan [21], and other polysaccharides from plants were effective in controlling blood sugar balance and regulating intestinal microbiota. NPs also play an essential role in the treatment of colitis-related diseases [22,23,24], and they are characterized by low toxicity, immunity enhancement, and prebiotic properties, in addition to facilitating IBD remission [25]. Although relatively few clinical trials have explored the use of NPs in IBD treatment, their effects under animal experiments, including in IL-10 deficient, dinitrobenzene sulfonic acid (DNBS), trinitrobenzene sulfonic acid (TNBS), and dextran sulfate sodium salt (DSS)-induced IBD mice models, have been investigated extensively [26]. According to the results of those pharmacological studies, NPs could ameliorate IBD symptoms through the activation of inflammation-related pathways, regulation of intestinal microbiota, modulation of the immune response, and repair of colon ulcer surfaces [27].

Interactions among intestinal microbiota, intestinal mucosal barrier, and immune system systems are thought to drive the pathological process of IBD; in particular, microbiota dysbiosis is a crucial factor inducing IBD [28]. Host–microbe imbalance can lead to abnormal inflammatory responses, and thereafter, excessive and uncontrolled immune responses induce imbalance in intestinal immune function [27]. In addition, intestinal epithelial barrier dysfunction is also associated with intestinal dysbiosis and intestinal mucosa inflammation. Therefore, this review aimed to summarize the impacts of NPs on IBD prevention and treatment in view of the extensive relationships among intestinal microbiota and immune function, intestinal barrier, and inflammation. Conclusions would offer insights in facilitating innovation and research on the underlying mechanisms of IBD.

## 2. Overview of Role of Intestinal Microbiota in IBD

Intestinal microbiota are complex microbial communities composed of trillions of fungi, viruses, archaea, microeukaryotes, and especially bacteria. Although most of them are beneficial microorganisms, some are pathogens. Intestinal microbiota significantly affect human and animal health [29,30]. The mechanisms involved in intestinal microbiota alleviation IBD are summarized in Figure 1. In IBD patients, increased populations of pathogenic bacteria in the intestine lead to imbalance in intestinal microbiota composition. The pathogenic bacteria migrating through damaged thin mucus layers cannot be protected by the impaired intestinal immune system. In addition, short-chain fatty acid (SCFA) and lactic acid production could be affected the gut environments. Consequently, the microbiota lose the essential source of energy for intestinal epithelial cells. Activated mucosal macrophages generate high levels of reactive oxygen species and inflammatory cytokines and thus exacerbate inflammatory responses. Conversely, in healthy intestine, intestinal microorganisms are distributed throughout the mucosal layer, and the intestinal epithelium is intact, so they are resistant to bacterial invasion. Intestinal microbiota and their metabolites, such as SCFAs, secreted proteins, indoles, organic acids, extracellular vesicles, and bacteriocins, have positive effects on the structural and functional integrity of the intestinal barrier, tight junctions, and mucosal immune response, which inhibits the growth of pathogenic microbes and facilitates homeostasis in the intestine.

## 3. NPs Impact on Intestinal Microbiota

Intestinal microbiota play essential roles in various physiological processes, such as inflammatory responses, immune system function, and energy homeostasis [31]. Over the past few decades, the impact of intestinal microbiota on host intestinal health has increasingly attracted the attention of researchers [32]. The human intestinal microbiota contains about 10^14^ bacterial cells; among them, Firmicutes are the most abundant, followed by Bacteroidetes [33,34]. Compared to healthy individuals, IBD patients have intestinal dysbiosis [35]. Dysbacteriosis, i.e., a decrease in microbial diversity, has been demonstrated in IBD diseases: for example, a decreased abundance of some beneficial bacteria, such as *Erysipelotrichales*, *Bactereroidales*, *Clostridial*, *Bifidobacterium*, and *Lachnospiraceae*, and increased abundance of some pathogenic bacteria, such as *Enterobacteriaceae*, *Pasteurellcaea*, *Veillonellaceae*, *Fusobacteriaceae*, Proteobacteria, *Ruminococcus gnavus*, and *Desulfovibrio* [25,27]. Specifically, *Enterorhabdus* spp., *Desulfovibrio* spp., *Alistipes* spp., and *Bacteroides* spp. are closely associated with protective effects against colitis [36]. Changes in such bacteria result in minimal resistance to the growth and colonization of pathogenic bacteria in the gut. Although pathogenic bacteria are relatively abundant in the intestines of IBD patients [37,38], there is no direct evidence of a specific pathogen that causes IBD.

NPs have been demonstrated to modulate the richness and diversity of intestinal microbiota and restore the structure of intestinal microbiota, resulting in significant alleviation of IBD symptoms [25,39]. As summarized in Table 1, polysaccharides from plants, fungi, animals, bacteria, and algae significantly regulated intestinal microbiota composition. For example, *Ziziphus jujuba Mill* var. polysaccharides significantly increased Firmicutes abundance and decreased Bacteroidetes abundance [40]. *Hericium erinaceus* polysaccharides can significantly increase microbial diversity in the gut of UC rats, with Verrucomicrobia, Firmicutes, Bacteroidetes, and Proteobacteria richness being close to levels in healthy rats [41]. In addition, *Glycyrrhiza* polysaccharides can regulate intestinal microbiota composition, with *Enterorhabdus* spp., *Odoribacter* spp., *Ruminococcaceae*_UCG_014, and *Ruminiclostridium* 5 as the potential targets [42]. Fucoidan could inhibit 1,2-dimethylhydrazine-induced *Prevotella* proliferation in rat colorectal cancerous intestine and increase *Alloprevotella* abundance [43]. *Sporisorium reilianum* polysaccharides significantly improved intestinal microbial diversity and microbiota composition in DSS-induced colitis. The abundances of Bacteroidota and Proteobacteria at the phylum level, and *Bacteroides*, *Coprobacillus*, *Streptococcus*, and *Lactobacillus*, at the genus level, returned to healthy levels as compared with normal control mice [44]. Water-soluble garlic polysaccharides alleviated DSS-induced colitis by modulating gut microbiota, and the key bacterial groups included *Muribaculaceae*, *Lachnospiraceae*, *Lachnospiraceae_NK4A136_group*, *Mucispirillum*, *Helicobacter*, *Ruminococcus_1*, and *Ruminiclostridium_5* [45].

## 4. Synergistic Effects between Intestinal Microbiota and Metabolites

The metabolism of complex carbohydrates is mainly mediated by carbohydrate-active enzymes (CAZymes) in the digestive tract. Intestinal microbiota encode a majority of CAZymes that are arranged in the polysaccharide utilization site gene cluster and are capable of degrading polysaccharides [34]. The degradation produces several functional metabolites, such as SCFAs, lactic acid, pyruvic acid, and ethanol, and gases, such as H_2_, CO_2_, CH_4_, and H_2_S [46,85]. Among them, SCFAs are the main fermentation products of NPs with biological activity that inhibit inflammation [50]. Specifically, the potential mechanisms of NPs that maintain intestinal homeostasis include a reduction in the secretion of pro-inflammatory factors and increase in functional metabolites through promoting the production of SCFAs, especially butyric acid. Butyric acid is an agonist of GPR41, GPR43, and GPR109a [86,87], and it can induce the differentiation of Treg cells and T cells by activating G-protein-coupled receptor signals in intestinal epithelial cells [88]. In addition, butyric acid could regulate the growth and function of dendritic cells and macrophages [89] as well as reduce the secretion of inflammatory cytokines [90,91] (Figure 2). In a previous study, *Tremella fuciformis* polysaccharides improved dysbiosis in DSS-induced UC mice and increased the concentrations of butyric acid, which stimulated FOXP3^+^ T cells [78] and upregulated the expression of *GPR41* and *GPR43* [53]. *Gracilaria lemaneiformis* polysaccharides could also promote the expression of SCFA receptors, GPR43 and GPR109a, and increase the concentrations of butyric acid as well as inhibit the secretion of pro-inflammatory factors, including IL-1β, IL-6, and TNF-α [92]. *Scutellaria baicalensis* polysaccharides also improved colitis by regulating Firmicutes and *Roseburia* abundance, enhancing the concentration of butyric acid, and reducing the levels of IL-6, IL-1β, and TNF-α [65]. In an IL-10-deficient mouse model, Goji berry polysaccharides have been demonstrated to alleviate inflammation through increasing the abundance of butyric acid-producing *Lachnospiraceae* and *Ruminococcaceae* [93]. Acetic acid also exhibits anti-inflammatory effects. Polysaccharides from *Crataegus pinnatifida* could alleviate colitis by modulating intestinal microbiota, increasing acetic acid contents, and reducing inflammatory cytokine secretion [50]. *Flammulina velutipes* polysaccharides could restore the structure and abundance of intestinal microbiota and promote the growth of lactic acid-producing *Lactobacillus* and butyric acid-producing *Ruminococaceae* while reducing the abundance of *Enterococcus* and Bacteroidetes [80]. In addition, tryptophan-metabolizing *Lactobacillus* strains can reduce intestinal inflammation by activating aromatic hydrocarbon receptors, especially *Lactobacillus reuteri*, which regulate the transformation of intraepithelial CD4^+^ T helper cells into immunoregulatory T cells by activating the aromatic hydrocarbon receptors through indole derivatives [94,95,96]. *T. fuciformis* polysaccharides have been reported to protect the colon from inflammation by increasing the level of the tryptophan metabolite [78].

## 5. Interaction between Intestinal Microbiota and Intestinal Barrier

The intestinal barrier, including mechanical, immune, chemical, and biological barriers [25], constitute the “first line of defense” against pathogen invasion in the human body. Figure 3 summarizes the composition and function of the intestinal barrier. Repairing damaged intestinal barriers has become the primary goal of IBD treatment [97]. Intestinal dysbiosis influences the integrity of the intestinal barrier and induces IBD [98]. Under normal circumstances, the mechanical, chemical, and immune barriers can all be regulated by microbial metabolism [99]. However, intestinal microbiota dysbiosis, i.e., a decrease in beneficial microbes and an increase in harmful microbes, would destroy the intestinal barrier [100]. The growth of bacteria with strong mucus degradation ability, for example, *Ruminococcus torques*, would be increased. In contrast, the proportion of bacteria with weak mucus degradation ability, such as *Akkermansia muciniphila* (*A. muciniphila*), decreases significantly. The abundance of some probiotics, such as *Lactobacillus*, *Bifidobacterium*, and *Prevotella*, would reduce when the bacterial membrane barrier is destroyed. In healthy individuals, beneficial bacteria that can inhibit the adhesion and colonization of pathogenic bacteria to protect against damage to the intestinal barrier are distributed in the mucus layer [85,101].

NPs as prebiotics could repair the IBD-induced damage to intestinal barrier by regulating intestinal microbiota [102] and re-establish a suitable growth environment for probiotics in the intestine and simultaneously inhibit the growth of potential pathogenic bacteria [102,103,104]. *H. erinaceus* polysaccharides can increase the relative abundance of key microbiota, including *Clostridium* spp., *A. muciniphila*, and *Desulfovibrio* spp., and improve the intestinal microbiota imbalance triggered by colitis [105]. In an αIL-10R-induced colitis mouse model, NP could upregulate the relative abundance of Firmicutes and downregulate the relative abundance of *A. muciniphila* [106]. Treatment of DSS-induced colitis mice with water-soluble polysaccharides from burdock increased the abundance of Firmicutes, *Ruminococcaceae*, and *Lacetospiraceae*, and it reduced the abundance of Proteobacteria, *A. muciniphila*, *Staphylococcus*, and *Bacteroides* [46].

An imbalance in intestinal microbiota can cause intestinal mucosal barrier dysfunction, increase intestinal cell permeability [107], and inhibit tight junction (TJs) and adherent junction synthesis [108,109,110]. NPs could obviously upregulate the expression levels of TJs, including occludin, claudin, JAMs 1-3, cingulin, and connexins from the (zonula occludens (ZOs). For example, α-glucan from a marine fungus, *Phoma herbarum* YS4108, directly repaired DSS-induced intestinal mucosal damage by upregulating the expression of ZO-1 and claudin-1 in the colon while restoring the populations of associated intestinal microbiota, i.e., increasing the abundance of *Bacteroidetes*, and decreasing the abundance of Firmicutes, Proteobacteria, *Clostridiales*, and *Lachnospiraceae* [73]. Chitosan can promote the expression of TJ proteins such as claudin-1, occludin, and ZO-1, and increase *Blautia* and *Lactobacillus* populations, in turn enhancing intestinal barrier function [111]. Fuzhuan Brick Tea polysaccharides promoted the expression of occludin and ZO-1 [112]. *Dictyophora indusiata* polysaccharides could enhance mucin, claudin-1, occludin, and ZO-1 expression to improve tissue structure and intestinal integrity by increasing the abundance of beneficial bacteria such as *Lactobacillus* spp [79]. Similarly, Chinese Yam polysaccharides can enhance the expression of ZO-1, claudin-1, occludin, and connexin-43, and reduce the relative abundance of Firmicutes, *Alistipes*, and *Helicobacter*, while increasing the relative abundance of Bacteroidetes, *Muribaculum*, *Roseburia*, and *Dubosiella* [49].

Reduced MUC2 mRNA levels in the inflammatory bowel are also associated with increased intestinal permeability in IBD mice models [113]. *G. lemaneiformis* polysaccharides can alleviate DSS-induced colitis through regulation of the expression of MUC2, with a reduction in the relative abundance of *norank_f_Erysipelotrichaceae*, *nclassified_f_Family_XIII*, *Acetatifactor*, and *Defluviitaleaceae_UCG-011*, and increasing the relative abundance of Actinobacteria, *Corynebacterium_1*, *Enterorhabdus*, and *Yaniella*.

## 6. Effect on Intestinal Immunity and Inflammatory Responses

Intestinal dysbiosis could increase bacterial translocation, which stimulates the activation of antigen-presenting cells, such as dendritic cells and macrophages, subsequently inducing changes in T cell subsets. Among them, Th1 cells produce IFN-γ/TNF-α, and Th2 cells produce IL-6; Th17 cells secrete IL-17, and Treg cells reduce IL-10, causing pro-inflammatory responses and exacerbating colon tissue damage [114,115]. Treg cells, another subgroup of CD4^+^ T cells, participate in host immune responses by secreting cytokines, such as IL-4 and IL-10, to facilitate immune balance (Figure 1) [116]. Interactions between Th17 and Treg cells maintain intestinal immune balance, and IBD is induced when changes in specific intestinal microbiota lead to Th17/Treg imbalance [117], and such pro-inflammatory cytokines would certainly enhance intestinal inflammation. Current findings indicated the role of immune-regulatory effects of NPs by targeting intestinal microbiota [115]. *Schisandra chinensis* polysaccharides restore intestinal microbiota and reduce the secretion of pro-inflammatory factors (TNF-α, IL-6, and IL-17), in addition to decreasing the relative abundance of Firmicutes, *Verrucomicrobia*, *Lactobacillus*, and *Turicibacter*, and increasing the relative abundance of Bacteroidetes, Actinobacteria, *Desulfovibrio*, and *Alistipes* [64]. *Morus atropurpurea* polysaccharides could alleviate inflammation in DSS-induced colitis mice and reduce the abundance of Proteobacteria, *Prevotellaceae*, and *Staphylococcus*, while increasing the abundance of *Lactobacillaceae*, *Lachnospiraceae*, *Ruminococcaceae*, and *Lactobacillus* significantly [46]. In addition, *Chrysanthemum morifolium* Ramat polysaccharides could reduce the abundance of opportunistic pathogens, *Escherichia* and *Enterococcus*, and inhibit the secretion of pro-inflammatory factors (IFN-γ, IL-6, IL-1β, and IL-17), while increasing *Clostridium*, *Butyricicoccus*, *Lactobacillus*, and *Bifidobacterium* abundance [118]. Exopolysaccharides from *Lactobacillus plantarum* NCU116 significantly reduced the amounts of CD11b^+^, CD45^+^, and CD3^+^ in DSS-induced colitis C57BL/6 mice. Meanwhile, the levels of *Lactobacillaceae* and *Bifidobacteriaceae* were increased significantly [119]. Similarly, *Ipomoea batatas* polysaccharides alleviated colonic inflammation by reducing the relative abundance of Proteobacteria, *Bacteroides*, and *Staphylococcus*, and increasing the relative abundance of *Lactobacillus*, *Roseburia*, and *Bifidobacterium*, when the secretion of pro-inflammatory factors (IL-1β, IL-6, and TNF-α) was inhibited [120]. *D. indusiata* polysaccharides reduced the production of pro-inflammatory cytokines (TNF-α, IL-1β, IL-6, and IFN-γ) in DSS-induced mice and the relative abundance of pathogenic bacteria (*Proteus*, *Enterobacteriaceae*, *Bacteroides*), and increased the relative abundance of *Lactobacillus* [121]. *Ramulus mori* polysaccharides inhibited the secretion of inflammatory cytokines IFN-γ and IL-6 and regulated intestinal microbiota abundance, including that of Firmicutes, *Bacteroides*, *Myxospirillum*, and *Paraprevotella*, and therefore, alleviated DSS-induced colitis in a mouse model [63].

The activation of inflammatory signaling pathways induces the production of pro-inflammatory cytokines, leading to the aggravation of IBD. NPs can improve intestinal inflammation by modulating TLR4, MyD88, and NF-κB signaling pathways (Figure 2) [122,123]. Polysaccharides from *Dendrobium officinum* prevented UC by decreasing the expression of TLR4 [124], restoring intestinal microbiota diversity, and increasing the abundance of *Bacteroides*, *Lactobacillus* and *Ruminococcus* while decreasing the abundance of Proteobacteria [53]. *Radix pseudostellaria* polysaccharides restored intestinal microbiota, upregulated TLR4 expression, increased the abundance of *Bacteroides* and *Pseudomonas,* and decreased the abundance of *Verrucomicrobia* and *Akkermansia* [39]. MyD88 is involved in protective inflammatory responses that regulate intestinal bacteria and the homeostasis of intestinal epithelial cells [125]. *Ganoderma lucidum* polysaccharides reduced the abundance of *Lachnoclostridium*, *Oscillibacter*, *Desulfovibrio*, *Alipipes*, and *Parasutterella*, and they inhibited the expression of MyD88, thereby alleviating DSS-induced colitis in mice [75]. Astragalus polysaccharides established intestinal microbiota balance by regulating the relative abundance of *Lactobacillus*, *Bifidobacteria*, and *Enterobacteriaceae*, and they inhibited the activation of the MyD88 signaling pathway so as to alleviate intestinal inflammation [126]. The NF-κB signaling pathway was activated by inhibiting the phosphorylation of NF-κB inhibitors, thereby promoting intestinal immune tolerance [127]. Polysaccharides from Chinese yams could inhibit the activation of the NF-κB signaling pathway and ameliorate DSS-induced microbiota imbalance by reducing *Alistipes* and *Helicobacter* abundance [49]. Polysaccharides from *S. baicalensis* Georgi ameliorated colitis via suppression of the NF-κB signaling pathway [128] while increasing the levels of Firmicutes, *Bifidobacterium*, *Lactobacillus*, and *Roseburia* and reducing the levels of *Bacteroides*, Proteobacteria, and *Staphylococcus* [65]. Similarly, Pacific abalone polysaccharides could inhibit the activation of the NF-κB signaling pathway, increase *Muribaculaceae* and *Lachnospiraceae* abundance, and decrease *Bacteroidaceae*, *Prevotellaceae*, and *Rickenellaceae* abundance [129].

## 7. Conclusions and Recommendations

Obviously, NPs were effective in IBD diseases, and their mechanisms were possibly related to the following three aspects: (1) repairing the damaged intestinal barrier, including mucosa, microbiota, intestinal permeability, etc.; (2) maintaining the intestinal microenvironment by modulating the diversity and richness of the intestinal microbiota community, for example, increasing the beneficial intestinal bacteria, improving bacterial translocation, and production of SCFAs; (3) ameliorating the severe immune responses of the host mucosa, such as promoting the secretion of anti-inflammatory cytokines and reducing the production of pro-inflammatory cytokines, activating epithelial lymphocytes and mucosal lamina propria immune cells, etc. In that way, they improve intestinal immunity and inflammatory responses in various IBD mice models.

Although NPs have been demonstrated to inhibit the activation of several inflammatory signaling pathways, no direct evidence of SCFAs produced by NPs fermentation with inflammatory responses or expression of peroxisome proliferator-activated receptor has been presented. In addition, glycan chain length and linkage type have been found to influence the immunogenicity and efficacy of glycoconjugate vaccines [130]; however, their structure–activity relationship in terms of IBD remains poorly understood. Furthermore, the bioavailability of NPs should be investigated to better understand their utilization in human body. Lastly, further large-scale clinical trials are required to facilitate the rigorous evaluation of their medical use.

## Figures and Tables

**Figure 1 foods-11-01084-f001:**
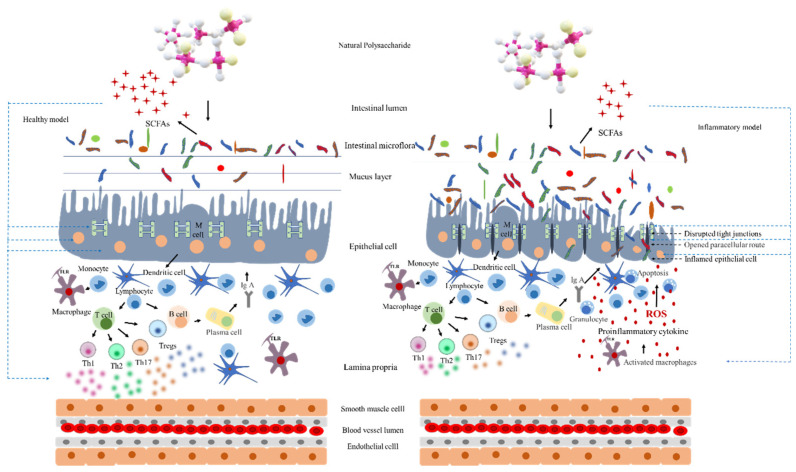
Schematic of mechanisms via which natural polysaccharides (NP) improve inflammatory bowel disease (IBD) through intestinal microbiota. A: Healthy model; B: Inflammatory model. ROS: reactive oxygen species.

**Figure 2 foods-11-01084-f002:**
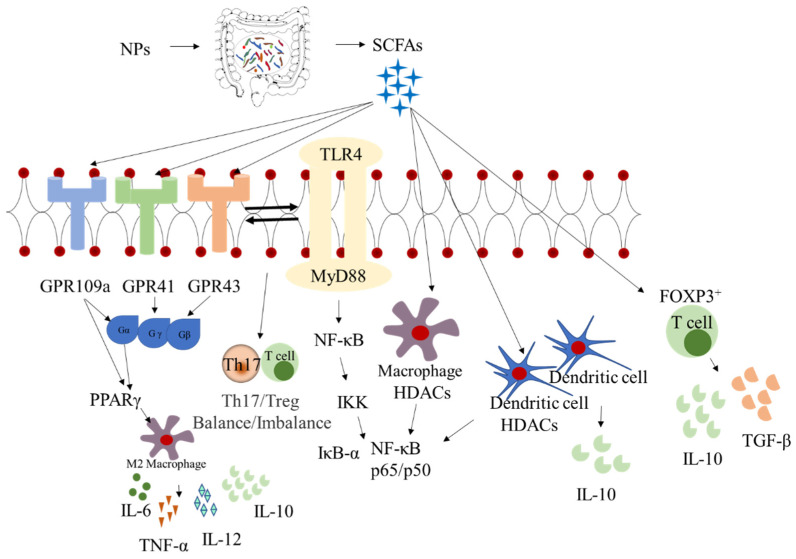
Effects of short-chain fatty acids (SCFAs) produced by intestinal microbiota on inflammatory bowel disease (IBD). HDAC: histone deacetylases; NF-κB: nuclear factor kappa B; GPR41: G protein coupled receptor 41; GPR43: G protein coupled receptor 43; GPR109a: G protein-coupled receptor 109 a; PPAR: peroxisome proliferator-activated receptors; TLR 4: Toll-like receptor 4.

**Figure 3 foods-11-01084-f003:**
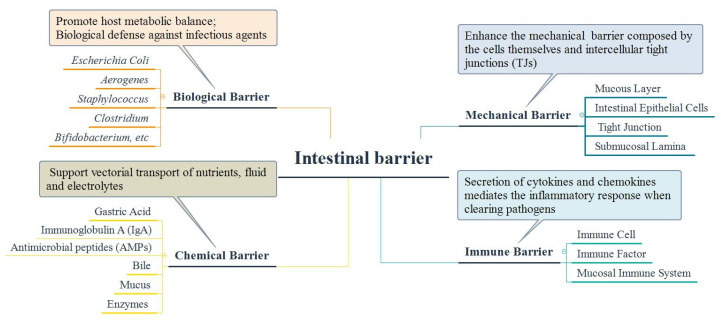
Composition and functions of the intestinal barrier.

**Table 1 foods-11-01084-t001:** Effects of natural polysaccharides on intestinal microbiota modulation in inflammatory bowel disease (IBD) models.

Natural Source	Intestinal Microbiota Modulation	Changes in SCFAs and LA	References
Plant			
*Allium sativum* L.	↑*Lactobacillaceae* *↓Lachnospiraceae*, *↓Muribaculaceae*, *↓Marinifilaceae*	↑AA*>* PA*>* IBA*>* IVA	[45]
*Arctium lappa*	↑Firmicutes, *↑Ruminococcaceae*, *↑Lachnospiraceae*, *↑Lactobacillus* *↓*Proteobacteria, *↓Alcaligenaceae*, *↓Staphylococcus*, *↓Bacteroidetes*	-	[46]
*Atractylodes macrocephala Koidz*	↑*Butyricicoccus*, *↑Lactobacillus* ↓*Actinobacteria*, *↓Akkermansia*, *↓Anaeroplasma*, *↓Bifidobacterium*, *↓Erysipelatoclostridium*, *↓Faecalibaculum*, *↓Parasutterella*, *↓Parvibacter*, *↓Tenericutes*, *↓Verrucomicrobia*	↑PA	[47]
*Camellia sinensis* L.	↑*Bacteroidaceae*, *↑Prevotellaceae*	↑AA*>* PA*>* BA	[48]
Chinese yam	*↓Alistipes*, *↓Helicobacter*, *↓Enterobacteriaceae*	-	[49]
*Crataegus pinnatifida*	↑*Alistipes*, ↑*Odoribacter*	↑AA	[50]
*Cyclocarya Palinurus*	↑*Lactobacillus*, *↑Lactobacillaceae*, ↓*Streptococcaceae*	↑AA*>* PA*>* BA*>* VA	[51]
*Dendrobium fimbriatum Hook*	*↑Romboutsia*, *↑Lactobacillus*, *↑Odoribacter* *↓Parasutterella*, *↓Burkholderia,* *↓Caballeronia,* *↓Paraburkholderia*, *↓Acinetobacter*	↑AA*>* PA*>* BA	[52]
*Dendrobium officinale*	*↑Bacteroides*, *↑Lactobacillus*, *↑Ruminococcaceae* *↓Proteobacteria*	↑AA*>* IBA	[53]
*Diospyros kaki* L.	↑*Lactobacillus*	↑PA*>* BA	[54]
*Durio zibethinus* Murr rind	↑*Lachnospiraceae* *NK4A136 group*	↑AA*>* BA*>* PA	[55]
*Ficus carica*	↑*S24-7*, *↑Bacteroides*, *↑Coprococus* *↓Escherichia*, *↓Clostridium*	↑AA*>* BA	[56]
Fructan	↑*Lactobacilli*, *↑Bifidobacteria*	↑PA*>* LA	[57]
*Fuzhuan brick tea*	↑*Bacteroides*, *↑Parasutterella*, *↑Collinsella*	-	[58]
*Lonicera japonica Thunb*	*↑Bifidobacterium*, *↑Lactobacilli* ↓*Escherichia coli*, *↓Enterococcus*	-	[59]
*Lycium barbarum*	↓*Enterococcaceae* ↑*Deferribacteraceae*	↑AA*>* BA*>* VA	[60]
*Morinda citrifolia* L.	↑*Dubosiella*, *↑Muribaculaceae*, *↑Ruminococcaceae_UGG-014,* *↑Ruminococcus_1* *↓Campylobacter*, *↓Escherichia-Shigella*, *↓Bilophila*, *↓Ochrobactrum*, *↓Vibrio*	↑AA*>* PA*>* BA	[61]
*Pseudostellaria* *heterophylla*	↑*Bacteroides*, *↑Pseudomonas* *↓Verrucomicrobia*, *↓Akkermansia*	↑AA*>* PA*>* BA	[39]
Purple sweet potato	*↑Bifidobacterium*, *↑Lactobacillus*, *↑Roseburia* ↓*Bacteroides*, *↓Proteobacteria*	↑AA*>* PA	[62]
*Ramulus mori*	↑*Clostridium XIVa*, *↑Mucispirillum*, *↑Paraprevotella*	↑AA*>* PA*>* BA	[63]
*Schisandra chinensis*	*↑Norank_f_Bacteroidales_S24-7_group*, *↑Desulfovibrio*, *↑Alistipes* *↓Lactobacillus*, *↓Turicibacter*	↑PA*>* BA*>* VA	[64]
*Scutellaria baicalensis* *Georgi.*	↑*Firmicutes*, *↑Bifidobacterium*, *↑Lactobacillus*, *↑Roseburia*	↑AA*>* PA*>* BA	[65]
Soybean	↑*Bifidobacterium*, *↑Lactobacillus*	↑AA*>* LA*>* BA*>* PA*>*	[66]
Xylan (corn cobs)	*↓Oscillibacter*, *↓Ruminococcaceae UGC-009*, *↓Erysipdatoclostridium*, *↓Defluviitaleaceae UCG-01*	↑BA	[67]
*Zizyphus jujuba cv. Muzao*	↑*Bifidobacterium*, *↑Bacteroides*, *↑Lactobacillus*	↑AA	[68]
*Ziziphus jujuba Mill.*	*↓Firmicutes* *↑Bacteroidetes*	-	[69]
Animal			
*Stichopus chloronotus*	↑*Megamonas*, *↑Bacteroides*, *↑Fusobacterium*, *↑Parabacteroides*, *↑Prevotella*, *↑Faecalibacterium*	↑AA*>* BA*>* IVA	[70]
Sea cucumber	↑*Parabacteroides distasonis*	↑BA	[71]
Oyster	↑*Akkermansia*	↑PA*>* BA	[72]
Fungus			
*Marine fungus* *Phoma herbarum* *YS4108*	*↑Bacteroidetes* *↓Firmicutes*, *↓Proteobacteria*, *↓Clostridiales*, *↓Lachnospiraceae*	↑BA*>* IVA	[73]
*Auricularia auricular-judae*	*↑Bacteroidetes* *↓Firmicutes*, *↓Ruminococcus*, *↓Deferribacteres*, *↓Actinobacteria*	-	[74]
*Ganoderma lucidum*	↑*Allobaculum*, *↑Bifidobacterium*, *↑Christensenellaceae R-7*, *↑Choerinum,* ↑*Lactobacillus,* *↑Firmicutes*, *↑Paraprevotella*, *↑Ruminococcus_1*, *↑Fusicatenibacter*, *↑Ruminiclostridium_5*, *↑Clostridiales* *↓Proteobacteria*, *↓Escherichia-Shigella*, *↓Barnesiella*, *↓Anaerotruncus*, *↓Tyzzerella*	↑AA*>* PA*>* BA	[75,76,77]
*Tremella fuciformis*	*↑Lactobacillus*, *↑Ruminococcaceae*, *↑Odoribacter*, *↑Helicobacter*, *↑Marinifilaceae*	-	[78]
*Dictyophora indusiata*	*↑Lactobacillus* *↓Proteobacteria*, *↓Gammaproteobacteria*, *↓Bacteroides*, *↓Bacteroidaceae*, *↓Enterobacteriaceae*	-	[79]
*Flammuliana velutipes*	*↑Ruminal butyrivibrios*, *↑Roseburia*, *↑Bacteroidales family S24-7*	↑BA*>* IVA*>* VA	[80]
*Hericium erinaceus*	↑*Ruminococcaceae*, *↑Allobaculum*, *↑Desulfovibrionales*	↑AA*>* BA	[41]
Bacteria			
*Lactobacillus* *planta-rum NCU116*	*↑Lactobacillaceae*, *↑Bifidobacteriaceae*	↑LA	[81]
Algae			
*Enteromorpha*	*↑Lactobacillus*	-	[82]
*Sargassum* *fusiforme*		-	[83]
*Porphyra haitanensis*	↑*Bacteroides thetaiotaomicron*, *↑Bacteroides ovatus*, *↑Defluviitalea saccharophila*, *↑Faecalibacterium prausnitzii*	↑AA*>* PA*>* BA	[84]

Note: AA (acetic acid), PA (propionic acid), BA (butyrate acid), IBA (isobutyric acid), IVA (isovaleric acid), VA (valeric acid), and LA (lactic acid); “-” means not mentioned in the reference; “↑” means “significantly increased the relative abundance of bacteria or productions of SCFAs”; “↓” means “significantly decreased the relative abundance of bacteria or productions of SCFAs”.

## Data Availability

No new data were created or analyzed in this study. Data sharing is not applicable to this article.

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
