# Peer review of "Influence of Natural Polysaccharides on Intestinal Microbiota in Inflammatory Bowel Diseases: An Overview"

_foods, 2022, doi:10.3390/foods11081084_

Round 1

Reviewer 1 Report

This paper deals with the topic of natural polysaccharides and its Influence on intestinal microbiota in inflammatory bowel diseases. The topic of the study is interesting and undoubtedly brings useful information in the knowledge of this area. The manuscript is well structured, but some sections need improvement.

I think it would be more useful to divide Table 1 into categories according to the different sources of polysaccharides (such as plants, fungi, bacteria, algae, etc.)

It would also be useful to mention in the text the various forms of polysaccharides (such as β-glucans) and their biological and / or therapeutic effects in humans.

Reviewer 2 Report

This review manuscript focuses on the gut microbiota and its fermentation products, mainly SCFAs, thought to mediate the effect of NPs in IBD. The gut microbiota exerts mitigating effects through multiple mechanisms, such as enhancing gut immunity, anti-inflammatory activity, and gut barrier protection through microbiota-dependent and microbiota-independent strategies.

The overall manuscript writing is fluent and concise, and the literature citation and discussion levels are clear and distinct, which has an excellent academic reference value for IBD research topics. In particular, the tabular literature organization and the cartoon schematic illustrations are beneficial to clarify the biological effects of NPs on IBD. Due to the rich polysaccharide literature compiled in this manuscript, it is a pity that the manuscript does not discuss the relationship between the source, structure and anti-inflammatory activity of polysaccharides and the production of SCFAs. Below are detailed comments:

  1. Briefly explain the definition of "natural polysaccharides (NPs)" in "Introduction", maybe refer to - https://doi.org/10.1016/B978-0-08-102553-6.00001-5.
  2. In Table 1, the research literature on natural polysaccharides is classified by plants, fungi, animals, algae, etc.
  3. Figures 1-3 are great. However, Figures 1-3 are not refferenced in the "6. Effect on intestinal immunity and inflammatory responses" text. It is recommended to make links to Figures 1-3 where appropriate.
  4. The overall “Conclusion” context is thin and fails to highlight the primary mechanisms by which polysaccharides improve IBD, such as (i) repairing the damaged gut barrier; (ii) adjusting the community diversity and richness of gut microbiota to optimistic criteria ; (iii) ameliorated the severe immune response of the host mucosa. It is recommended to supplement the critical points of the conclusions.
  5. The footnotes in Table 1 can be further improved.
